## PERSPECTIVE

# Single electron transfer-based peptide/protein bioconjugations driven by biocompatible energy input

Yue Weng[1], Chunlan Song[2], Chien-Wei Chiang[2✉] & Aiwen Lei [2✉]

Bioconjugation reactions play a central facilitating role in engendering modified peptides and proteins. Early progress in this area was inhibited by challenges such as the limited range of substrates and the relatively poor biocompatibility of bioconjugation reagents. However, the recent developments in visible-light induced photoredox catalysis and electrochemical catalysis reactions have permitted significant novel reactivities to be developed in the field of synthetic and bioconjugation chemistry. This perspective describes recent advances in the use of biocompatible energy input for the modification of peptides and proteins mainly, via the single electron transfer (SET) process, as well as key future developments in this area.

The growing interest in peptides as drug candidates, and for use in preparing antibody–drug conjugates in current therapeutics has stimulated increased interest in new bioconjugation strategies[1,2]. Introducing new methodologies to discover other types of peptide and protein modifications have become important and attractive to researchers[3–7]. Indeed, methods were previously available for labeling and modifying the amino acid residues of peptides and proteins. However, developing more methods that are specific for individual amino acids promises to allow scientists in the areas of the chemical biology, the life sciences, and clinical medicine to apply these methods for specific purposes[3–13]. This concept, for example, is embodied in the recent, efficient Pd-mediated methods for the arylation of cysteine reported by Buchwald and Pentelute[14,15]. In addition, bioconjugation methods targeting poorly nucleophilic, less surface-exposed hydrophobic amino acid residues have also attracted the attention of investigators in this field. Areas such as the bioconjugation of methionine through redox reactivity have been pursued by Toste and Chang[16], as well as Gaunt[17].

During the past decades, traditional methods for labeling amino acid residues required the introduction of reactive reagents for relative unreactive amino acids or to employ electrophiles that are reactive with respect to cysteine (Cys) or lysine (Lys)[10,11,18,19]. While active labeling reagents have now been added to biomolecule systems, issues related to their selectivity, toxicity, and biocompatibility remain a concern to scientists. In addition, common sense tells us that

---

[1] Ministry-of-Education Key Laboratory for the Synthesis and Application of Organic Functional Molecule & School of Chemistry and Chemical Engineering, Hubei University, 430062 Wuhan, Hubei, China. [2] College of Chemistry and Molecular Sciences, Wuhan University, 430072 Wuhan, Hubei, China. ✉email: cwchiang@whu.edu.cn; aiwenlei@whu.edu.cn

cysteine and lysine are the most useful bioconjugation handles because of their high reactivity. The photoredox/electrochemical process would confront the issue of side reactions that can occur at highly reactive Cys and Lys units, which would be the most important aspect, as well as the case of polar reactions[20]. In some examples of photoredox bioconjugation, side reactions such as the oxidation of Cys units can occur, due to the fact that the oxidative nature of excited photosensitizer is overly strong[21]. However, choosing a suitable photosensitizer based on its redox potential is essential, since this would minimize the problem of side reactions. In addition, in some examples, labeling cysteine and lysine, for example, continued to have limitations related to site- and chemo-selectivity, as well as the functional group tolerance of the substrates[10].

In the research fields of bioconjugation and bioorthogonal chemistry, site-specific ligation methods are clearly still one of the biggest challenges. Previously, traditional reagents can be used in some site-specific ligation reactions that involve cysteine and lysine, leading to improved pharmacological properties of a homogeneous population of proteins/peptides conjugates. However, bioconjugation strategies that involve mild and biocompatible reaction conditions (i.e., room temperature, atmospheric pressure, physiological pH, and aqueous buffer solutions as solvent), thus resulting in the formation of stable bioconjugates have yet to be clearly exploited for most amino acids[22,23].

In the field of synthetic chemistry, the application of a single electron transfer (SET) strategy for constructing new complicated molecules have recently become more popular. By introducing methods based on electrochemical synthesis or visible-light-induced photoredox catalysis for generating molecules, new breakthroughs compared to a traditional molecular synthesis approach might be possible. Because of their unique nature, electrochemical synthesis and photoredox catalysis methods could enable unprecedented transformations, which explains the increasing attention they have received from the scientific community[24–28]. Here, we highlight some examples of visible-light-induced photoredox catalytic and electrochemical synthetic bioconjugation reactions.

Electrochemical synthesis or visible-light-induced photoredox approaches, which involve a SET with an organic substrate acting as either an electron donor or acceptor, involving the formation of radical intermediates would provide mild reaction conditions, leading to the generation of target products with less side reactions. Therefore, due to their mild reaction conditions, high site-selectivity and high functional group tolerance, these techniques represent an attractive solution for achieving the next generation of biomolecular modifications. Mastering such a powerful strategy means no less than finding a method that can be used to activate amino acid residues, ideally with a high site-/chemo-selectivity (Fig. 1).

## Photoredox catalytic bioconjugations

The reason for applying visible-light-induced photoredox catalysis to the development of novel methodologies for bioconjugation is apparent. Opposed to the modification of a peptide/protein by UV irradiation, which would disrupt the conformational integrity of a protein, visible-light-induced photoredox methods would preserve the structures of such bioactive molecules[29]. In addition, photoredox bioconjugation reactions could be performed under mild and biocompatible conditions, where the kinetics of the reaction could be relatively easily controlled. Thus, the photocatalytic modification of a single amino acid in a peptide or a protein can now be achieved. Visible-light-induced photoredox methods for modifying aromatic amino acid residues, nonaromatic amino acid residues, and side chains

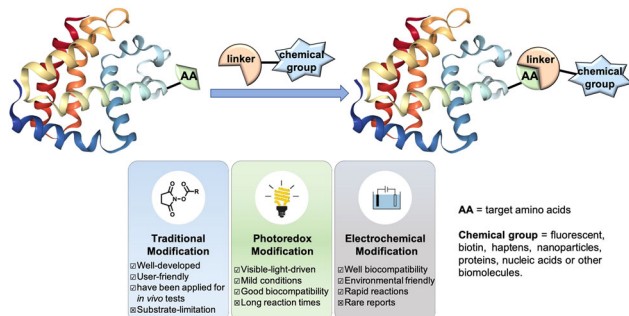

**Fig. 1 Recent bioconjugation strategies.** Comparison of properties between traditional modification, photoredox modification, and electrochemical modification of biomolecules.

or C-terminal carboxylic acid residues have recently been reported[30–32]. However, in comparison to the traditional bioconjugation approaches, visible-light-induced photocatalytic methods are still quite rare.

**Aromatic amino acids (Tyr and Trp).** Regarding recent bioconjugation methods, the most attractive targets for new synthetic transformations have focused on aromatic amino acids. Because of the C($sp^2$)–H functionalization of aromatic compounds has been extensively developed recently[24,26], chemists have turned their focus on the late-stage functionalization of the bioactive compounds and bioconjugation/bioorthogonal chemistry[14]. Currently, some photoredox catalysis and electrochemical synthesis approaches are being applied to examine the possibility of modifying aromatic amino acid residues in biomolecules. These reports have focused on the modification of tyrosine and tryptophan, because their reactivity can be activated by means of a SET strategy. On the other hand, due to the difficulty associated with the activation of histidine and phenylalanine, these types of bioconjugation methods were still rare in the literature[6,10] and their photocatalytic modification remains an unmet challenge.

*Tyrosine.* Tyrosine, an important amino acid, is found in nearly all proteins and polypeptides, such as tyrosine protein kinases, kisspeptin, and myoglobin. Because of its low abundance in native proteins and the fact that it is rarely found on protein surfaces, tyrosine is considered to be a site-selective modification target[12,33,34]. It is worth noting that the tyrosine residue may exist in either the phenol or phenolate form; while the latter is more reactive, at physiological conditions, the phenolate form is a minor contributor in most cases. Therefore, introducing photoredox catalysis for modifying tyrosine holds some promise. A few studies of photoredox bioconjugation specifically targeting tyrosine (Tyr) have recently been developed.

In 1999, Kodadek and coworkers reported on the earliest photocatalytic modification of proteins (Fig. 2a). They developed a cross-linking reaction that was initiated by the generation of a tyrosyl radical, which was then coupled with other nucleophiles[21,35]. In this case, the de novo achievement of this process involved the photolysis of a tris-bipyridyl Ru(II) complex with visible light in the presence of an electron acceptor, i.e., ammonium persulfate. High yields of cross-linked products can be obtained rapidly.

Parish, Krska, and coworkers reported in the development of a photocatalytic method for the trifluoromethylation of tyrosine side chains under biocompatible conditions (Fig. 2b). The derivatized tyrosine was prepared using Ir[dF(CF$_3$) ppy]$_2$(dtbbpy)PF$_6$ (dF(CF$_3$)ppy = 3,5-difluoro-2-[5-(trifluoro-methyl)-2-pyridinyl-N]-phenyl-C) as the catalyst, with 20 eq. of

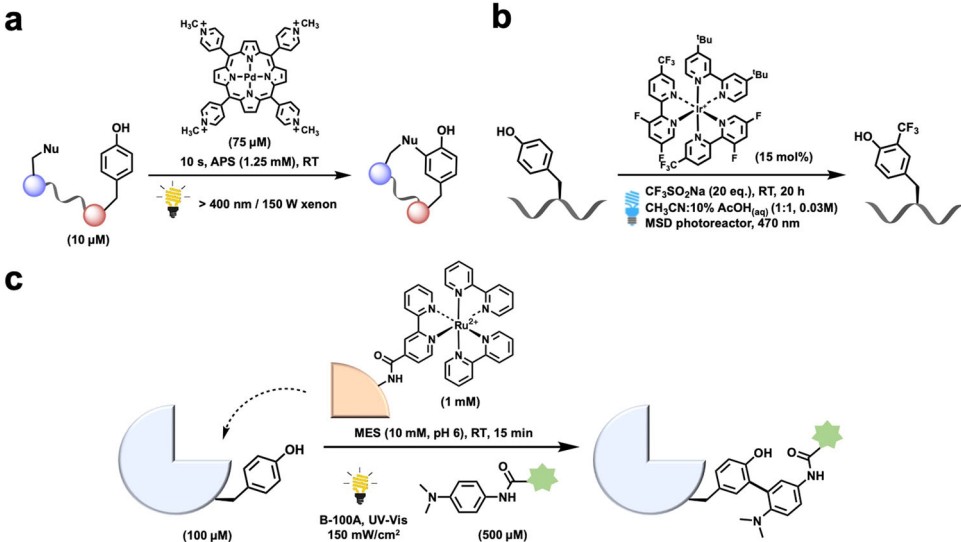

**Fig. 2 Photoredox Tyr modification. a** Cross-linking reaction by the generation of a tyrosyl radical, reported by Kodadek in 1999 (ref. [35]). **b** Photocatalytic trifluoromethylation of tyrosine, reported by Parish and Krska in 2018 (ref. [36]). **c** Ligand-directed protein photocatalysis for modifying tyrosine residues in proteins, reported by Nakamura in 2013 (ref. [38]).

$CF_3SO_2Na$ as the fluoride source. This method was quite selective toward tyrosine in the presence of histidine and phenylalanine residues[36].

Nakamura and coworkers also recently developed a photocatalytic method for modifying tyrosine residues in proteins (Fig. 2c). Using a ligand-directed protein photocatalyst, [Ru (bpy)$_3$]$^{2+}$, tyrosyl radicals could be generated upon light irradiation and then reacted with $N'$-acyl-$N,N$-dimethyl-1,4-phenylenediamine[37,38].

*Tryptophan*. Tryptophan (Trp) is the rarest of all proteinogenic amino acids with an abundance of ~1.3%. It is noteworthy, however, that ~90% of proteins contain at least one tryptophan residue[39,40]. Therefore, the chemoselective modification of tryptophan would allow site-selective bioconjugation on nearly all known endogenous substrates. By applying traditional methodologies, such as transition metal catalyzed C–H functionalization[41,42], as well as metal-free organoradical conjugation strategies[43], in the majority of methods for the modification of tryptophan, the C-2 position of indole was found to be the most reactive. However, traditional strategies for modifying tryptophan are still limited by their biocompatibilities. It is clear that innovative strategies for overcoming the remaining challenges would also greatly contribute to expanding the toolbox of synthetic methods for posttranslational modification methods. Therefore, in the field of chemical biology, the development of photoredox bioconjugation approaches for targeting tryptophan represent significant advancements.

In 2017, Chen and coworkers developed a photochemical C–H perfluoroalkylation reaction for labeling tryptophan residues in oligopeptides at the C-2 position (Fig. 3a)[44]. These researchers found that perfluoroalkyl iodides could be activated with tertiary amines; thus, creating a method that does not require an additional photoredox catalyst and employed direct irradiation from a compact fluorescence light or sunlight.

Lei, Chiang, and coworkers recently reported on the visible-light-induced photoredox trifluoromethylation of tyrosine (Fig. 3b)[45]. To date, the incorporation of fluorine into biomolecules is rare, but potential applications in structural reengineering and as reporter tags for $^{19}$F-NMR raise interesting possibilities[46]. In this method, the selective trifluoromethylation

of Trp residues in peptides with 2 eq. of $CF_3SO_2Na$ salt was accomplished, using Ir[dF(CF$_3$)ppy]$_2$(dtbbpy)PF$_6$ as the photocatalyst. With the direct incorporation of a trifluoromethyl group in a peptide, this direct trifluoromethylation strategy will allow studies of various fluorinated peptides via the use of $^{19}$F-NMR. In addition, the method was applied to the modification of relevant polypeptides, thus broadening the synthetic scope of the protocol for the assembly of a variety of Trp-containing peptides. Interestingly, based on EPR results, radical trapping experiments, and Stern–Volmer experiments, the author proposed that the visible-light-induced photoredox trifluoromethylation of Trp proceeded via a radical–radical cross-coupling pathway, instead of via a radical addition route, which was previously assumed.

Taylor and coworkers recently reported on a photochemical process for the selective modification of Trp residues in peptides, as well as small proteins (Fig. 3d). In this case, electron-responsive N-carbamoylpyridinium salts and UV-B light were employed to cleave the pyridinium N–N bond, and concomitantly transfer a carbamoyl group to a Trp residue. Through a photoinduced electron transfer between the Trp residue and the pyridinium salt, the reaction displayed an excellent selectivity for Trp, and, importantly, the reaction proceeds under pure aqueous conditions without the need for organic cosolvents and photocatalysts[47].

On the other hand, since the benzylic position of aromatic compounds is quite reactive, several benzylic C–H functionalization reactions using traditional synthetic chemistry have been reported (Fig. 3c)[24,48]. Therefore, employing this type of activation to functionalize aromatic amino acids would provide an opportunity to selectively modify biomolecules in an unprecedented manner. Shi and coworkers recently reported on photoredox chemoselective peptide conjugation at the β-position of tryptophan[49]. In this report, they proposed that the nitrogen in the indole moiety of tryptophan is oxidized by the excited state of the photocatalyst, and a radical cation species is then generated by irradiation with an integrated singleton photoreactor (450 nm). Because the benzylic proton is relatively acid in nature, the stabilized Trp-skatolyl radical species that was generated in the presence of $K_2HPO_4$ was involved to the catalytic cycle. Eventually, this radical undergoes bioconjugation with a Michael acceptor to afford the desired product. This transformation of the

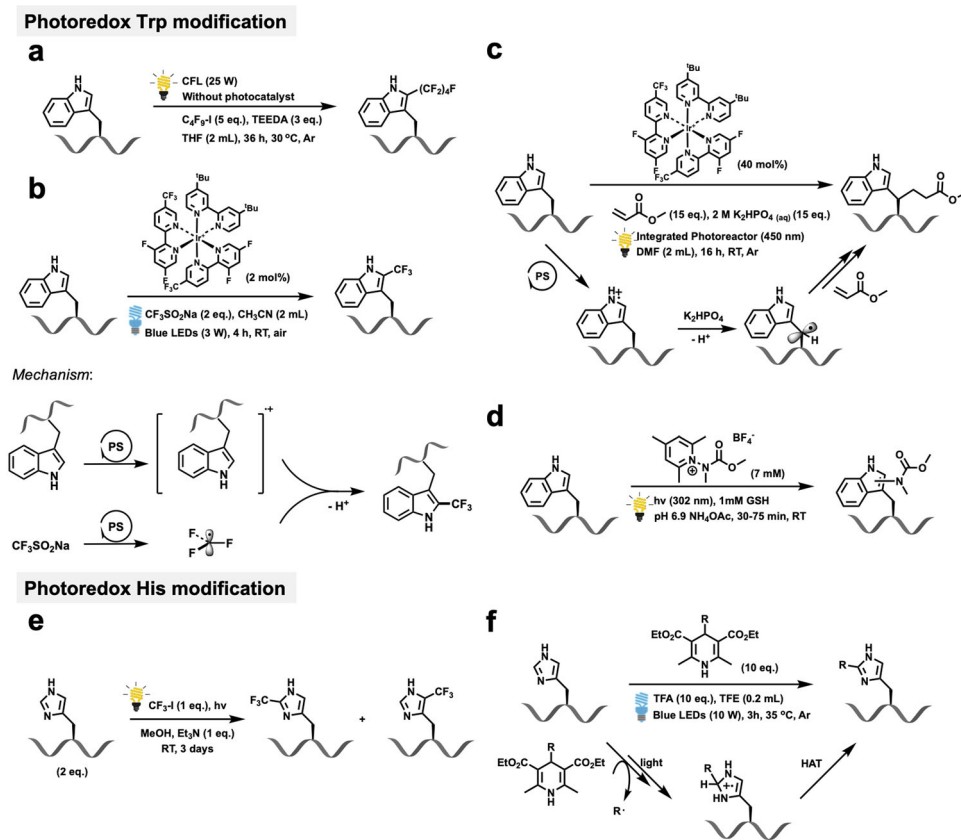

**Fig. 3 Photoredox Trp and His modifications, and related mechanistic studies. a** Photochemical C–H perfluoroalkylation for labeling tryptophan, reported by Chen in 2017 (ref. [44]). **b** Photoredox trifluoromethylation of tyrosine, reported by Lei and Chiang in 2019 (ref. [45]). **c** Photoredox tryptophan bioconjugation with at the β-position of tryptophan, reported by Shi in 2018 (ref. [49]). **d** Photochemical process for the selective modification of tryptophan residues with pyridinium salt, reported by Taylor in 2020 (ref. [47]). **e** Photochemical trifluoromethylation of histidine, reported by Cohen in 1990 (ref. [53]). **f** Histidine-specific peptide/protein modification via photoredox C–H alkylation reaction, reported by Wang and Chen in 2019 (ref. [54]).

benzylic position on an aromatic amino acid provides an alternative approach for the modification of peptides and proteins.

*Histidine*. Histidine is often found in the active site of enzymes and is of crucial importance in mechanisms that involve the abstraction or donation of a proton. In addition, it is the only residue with a p$K_a$ in the physiological range. However, issues associated with histidine bioconjugation include the p$K_a$, hydrogen bonding, as well as solvent exposure. There are relatively few conventional bioconjugation methods available for activating histidine. In addition to the traditional approaches that involve chemical functions, such as Michael addition and reactions with epoxides and diazonium salts[50–52], the SET-based photoredox methods promise to provide a new strategy for modifying histidine uner mild conditions and in a straightforward manner.

Early studies of the direct photochemical modification of histidine appeared in 1990. Cohen and coworkers reported on the photochemical trifluoromethylation of His in a tripeptide Glp-His-Pro-NH$_2$ (TRH) and produced two isomers, 2- and 4(5)-CF$_3$-Im-TRH (Fig. 3e). By introducing CF$_3$I as the source of the trifluoromethyl group, the tripeptide could be modified in MeOH under UV irradiation[53]. However, the chemoselective modification of histidine continues to remain a difficult challenge in bioconjugation chemistry.

Importantly, Wang, Chen, and coworkers recently developed a His-specific peptide/protein modification protocol that involved a visible-light-induced photoredox C–H alkylation reaction (Fig. 3f)[54]. In this case, peptide modification involved using

C$_4$-alkyl-1,4-dihydropyridine as the labeling reagent under irradiation by visible light. Through the Minisci-type reaction, this protocol exploits the electrophilic reactivity of histidine. Moreover, its utility has been demonstrated in a series of peptide drugs, natural products, and small proteins.

## Nonaromatic amino acids (Cys, Ser, and Thr)

*Cysteine*. In traditional bioconjugation chemistry, the cysteine residue is the most popular and convenient target for the post-translational modification of peptides and proteins because of its high nucleophilicity[3,10,14]. In addition, the relatively low natural abundance of cysteine residues in proteins would allow a specific site to be modified at a predetermined position[39]. Furthermore, because of its high nucleophilicity, particularly in the thiolate form, this allows a rapid and efficient modification a possibility.

In contrast, methionine, which also contains a sulfur atom in its residue, there are only a few examples of traditional methods for its modification in the literature and no usable photocatalytic methodologies for methionine bioconjugation have been reported[16,17,55,56]. That is because methionine has a fundamentally lower nucleophilicity and is highly hydrophobic[6]. However, while photoredox C–H functionalization was introduced to the late-stage application of a Met-containing peptide, C–H cyanation of the methyl group on methionine could be achieved directly[56].

On the other hand, since the thiol (thiolate) group of cysteine can readily react in SET reactions, the use of cysteine in conjunction with visible-light-induced photoredox bioconjugation

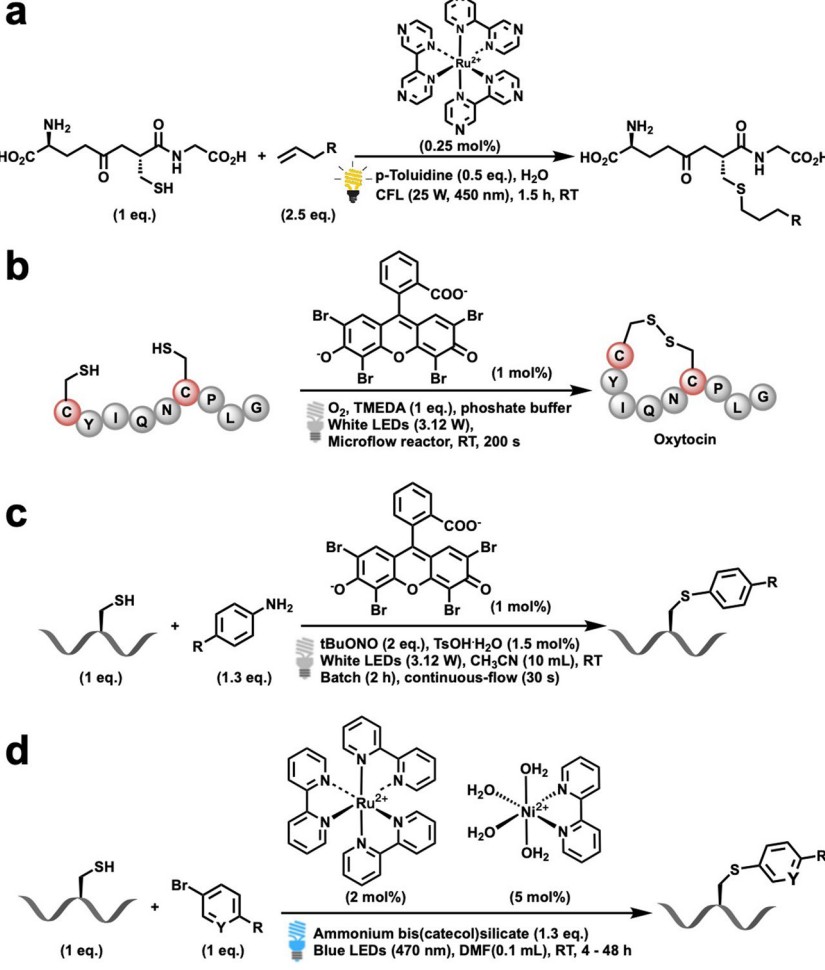

**Fig. 4 Recent examples of photocatalytic Cys modification. a** Photoredox thiol–ene reactions of cysteine-containing biomolecules, reported by Yoon in 2014 (ref. [57]). **b** Photoredox disulfide bonds generation between two cysteine, reported by Noël in 2015 (ref. [58]). **c** Photocatalytic arylation reaction of cysteine with aryldiazonium salt, reported by Noël in 2017 (ref. [60]). **d** Ni/photoredox dual catalytic system for the arylation of cysteine, reported by Molander in 2018 (ref. [63]).

systems are predicted to be important and valuable applications in this area of research.

In 2014, Yoon and coworkers developed a visible light photocatalytic method (Fig. 4a)[57]. Using [Ru(bpz)₃]²⁺ as a photocatalyst and *p*-toluidine as a redox mediator, they succeeded in coupling cysteine with various olefins. This protocol involved thiol–ene reactions of cysteine-containing biomolecules with the thiol component, as the limiting reagent under visible light irradiation. In this case, *p*-toluidine served as a redox mediator to catalyze the otherwise inefficient photooxidation of thiols, thus resulting in the development of a co-catalytic oxidative system.

Noël and coworkers recently reported several examples of the photocatalytic modification cysteine residues (Fig. 4b). They employed Eosin Y or TiO₂ as the photocatalyst to produce disulfide bonds between two cysteine residues[58,59]. It is well known that disulfides play an important role in protein folding, and the formation of secondary and tertiary structures. This method can be performed on deprotected peptides and showed a great tolerance for various sensitive amino acid residues. They also developed a continuous-flow protocol for use in the photocatalytic synthesis of oxytocin.

In addition, Noël and coworkers. developed a photocatalytic arylation reaction based on the Stadler–Ziegler reaction (Fig. 4c)[60]. On treatment with aryldiazonium salts, aryl radicals could be

generated in the presence of Eosin Y as the photocatalyst. The aryl radicals can subsequently be trapped by cysteine with the formation of a C–S bond, resulting in the ultimate modification of cysteine.

Importantly, the use of a combination of photoredox/transition metal dual catalysis methodologies was recently reported[61,62]. The dual catalysis strategy demonstrated a more directed and alternative reactivity in comparison to the simple SET pathway of photoredox catalysis. In 2018, Molander and coworkers reported on a Ni/photoredox dual catalytic method for the arylation of unprotected cysteine residues (Fig. 4d)[63]. This protocol employed ammonium bis(catechol)alkylsilicate as a hydrogen atom transfer (HAT) precursor and a photocatalyst. While the photocatalyst was excited by visible light, the HAT precursor could be activated to form a radical species, which then performed a rapid HAT process with cysteine. The photoredox-generated thiyl radical then became involved in the Ni-catalytic cycle to participate in a cross-coupling reaction with the aryl bromide. Notably, this reaction achieved scalability and a high throughput.

On the other hand, the reported photocatalytic desulfurization of cysteine also represented a useful conversion of cysteine bioconjugation. In 2016, Guo and coworkers explored the specific, visible-light-induced desulfurization of cysteine-containing peptides. Using Ru(bpy)₃²⁺ as a catalyst, the reaction

was found to proceed at room temperature in an aqueous/organic cosolvent. In this protocol, the desulfurization of cysteine residue was successfully applied to the synthesis of peptides through the ligation–desulfurization protocol[64]. In addition, Cabrele, Reiser, and coworkers developed a scalable protocol for the photocatalytic desulfurization of cysteine. In this study, they employed 0.01 mol% of Ir(dF(CF$_3$)ppy)$_2$(dtbbpy)PF$_6$ as a photocatalyst, with the treatment of triethylphosphite under biphasic reaction conditions, giving rise to the corresponding desulfurized product in high yields. For example, cystine, penicillamine, and reduced and oxidized glutathione could also be desulfurized by this method at a pH of ~7.0. Interestingly, while a Cys- and Met-containing decapeptide is employed, these sulfur-containing residues could be oxidized to disulfides and sulfoxides[65].

*Serine.* Under conditions of physiological pH, it is difficult to modify serine (Ser, S) and threonine (Thr, T) because of their poor nucleophilicities. Previously, studies indicated that Ser/Thr modification usually require rather harsh conditions, which may be incompatible with Cys, Tyr, Trp, and Met. Examples of traditional methods for the direct labeling of serine and threonine units are limited[6,66]. However, by employing a photoredox/Ni dual catalysis system, the challenges associated with serine modification could be overcome and C–O bond formation could be achieved. According to a report by Sciammetta, Lee, and coworkers, this successful cross-coupling reaction was carried out with both serine and threonine[67]. In this report, chemoselective C–O bond formation was achieved in intermolecular reactions between serine and an aryl bromide (Fig. 5a). This C–O coupling reaction also revealed the existence of the side chain-to-tail macrocyclization of peptides containing a β-turn motif. Based on this concept, recent advances in the field of C–H activation through the use of a combination of photoredox/transition metal dual catalysis might also provide opportunities to reach the previously thought to be unreactive amino acid residues. Kanai and Oisaki reported on a strategy for the photoredox α-C–H alkylation of serine containing peptides[68]. The reaction of serine residues with vinyl diethyl phosphonate was accomplished by the use of an electron-deficient borinic acid–ethanolamine complex with a photocatalyst and a HAT catalyst under conditions of irradiation, with blue LEDs. When a protected amino acid or Ser-containing dipeptide was used, the alkylation reaction proceeded successfully. This result represents the potential utility for the late-stage modification of Ser-containing peptides.

**Side chains or C-terminal carboxylic acids.** In 2016, MacMillan and coworkers reported on a visible-light-induced photoredox bioconjugation reaction for C-terminal carboxylates of polypeptides, which also proceeded through a SET pathway[69,70]. Since naturally abundant carboxylic acids can function as carbon-centered radicals, a transient radical intermediate was generated, with the subsequent elimination of $CO_2$ (Fig. 5b). The radical intermediate then underwent coupling with diethyl ethylidenemalonate, which served as the electrophilic partner, as well as a Michael acceptor. In addition, to widening the applications of this method and increasing biological compatibility, the authors employed riboflavin tetrabutyrate as an efficient water-soluble photocatalyst. In this report, the photoredox bioconjugation method can be applied to endogenous peptides and was successfully demonstrated on a small protein, namely, human insulin. The decarboxylation of the C-terminus that occurred selectively at the chain A was alkylated using this method.

Moreover, Opatz and coworkers reported on a photoredox alkenylation of carboxylic acids containing peptides in 2019 (ref. [71]). In this report, they applied the photoredox alkenylation method, in which carboxylic acids are used to modify peptides. By introducing 1,4-dicyanoanthracene as an organic photoredox catalyst, the carboxylic acid underwent oxidative decarboxylation, with the subsequent generation of vinyl sulfone and α,β-unsaturated nitrile substituted peptides.

## Electrochemical bioconjugation methods

The toxicity and biocompatibility of traditional organic chemistry and photosensitizer-caused contamination associated with photoredox chemistry on bioconjugation reactions continue to be issues. The use of electrochemical synthesis in bioconjugation and chemical biology promises to provide milder, biocompatible, and environmentally friendly conditions for constructing the novel types of bioconjugation with unreactive substrates[26,27]. In addition, because both the operating current and potential can be controlled, as well as the novel reactivity associated with electrochemical bioconjugation, modifying large biomolecules, such as proteins and antibodies, promises to be more straightforward than photoredox catalysis and other traditional methods. In other words, electrosynthesis techniques could eliminate the biocompatible issue associated with adding a reactive substrate, as well as a metallo-catalyst[28].

The electrochemical modification of peptides, proteins, and antibodies were previously limited to direct electrolysis for

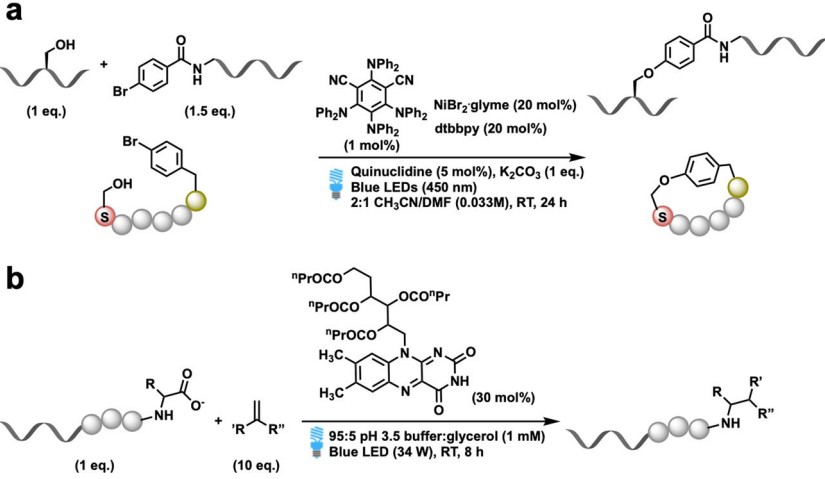

**Fig. 5 Photocatalytic Ser modification and C-terminal modification. a** Photoredox/Ni dual catalytic serine modification, reported by Sciammetta and Lee in 2019 (ref. [67]). **b** Photoredox bioconjugation through the decarboxylation of C-terminal carboxylates of polypeptides, reported by MacMillan in 2017 (ref. [69]).

cysteine–cystine interconversions[72], site-selective cleavage[73], and oxidative functionalization[74]. To date, only a few studies have appeared concerning the electrochemical activation of biomolecules. Due to the operational simplicity and inherent flexibility of the direct modification of proteinogenic amino acids, electrochemically chemo-specific modification promises to become a powerful strategy for applications in chemical biology, medical chemistry, and clinical pharmacology research. As shown by the examples described herein, electrocatalytic approaches offer the possibility of carrying out diverse chemical modifications of amino acids under mild and, in some cases, biocompatible reaction conditions.

**Tyrosine.** The use of an electrochemical bioconjugation systems has some advantages regarding tyrosine as a labeling target, because (1) the content of tyrosine in most native proteins is low and it is not usually found on protein surfaces; (2) the reactivity of tyrosyl species can be tuned by selective deprotonation.

One of the electrocatalytic modifications of Tyr was reported by Gouin, Boujtita, and coworkers[75]. They reported on a traceless electrochemical method for protein bioconjugation. The electrochemically promoted tyrosine-click (e-Y-CLICK) allowed the chemoselective Y-modification of peptides and proteins with labeled urazoles (Fig. 6a). Using this method, only a low potential (0.36 V) was needed and the urazole anchors could be activated in situ without oxidizing the sensitive amino acids in the protein. The protocols were successfully performed in pure aqueous buffers and cosolvents, a scavenger or an oxidizing agent were not necessary. Furthermore, the chemo-selectivity was also such that lysine modification due to PTAD decomposition, double Tyr modifications, and the oxidation of

thiols was not observed. Importantly, as a complement to the current arsenal of techniques, the e-Y-CLICK protocols could be successfully used in the electrosynthesis of relevant peptides and proteins, such as oxytocin, angiotensin, serum bovine albumin, and epratuzumab.

In addition, Nakamura, Sato, and coworkers reported on electrochemically activated SET-modified surface-exposed tyrosine residues of proteins[76]. In this study, selective protein modification could be achieved by targeting surface-exposed tyrosine residues. Under the horseradish peroxidase (HRP)-catalyzed SET and an electrochemically activated Tyr click reaction, N-methylated luminol derivative can be attached to Tyr residues of proteins and antibodies. The author also subjected CDR-modified antibodies to in vivo imaging and antibody–drug conjugated (ADC).

Lei, Chiang, and coworkers recently developed an electrochemically promoted labeling strategy for modifying tyrosine-containing biomolecules with phenothiazine (PTZ) derivatives using simple, mild, and clean conditions[77]. Investigations into the mechanism indicated that the reaction proceeded through the single-electron oxidation of PTZ to generate a nitrogen radical cation, which, when added to the *ortho*-position of phenol, resulted in the modification of tyrosine (Fig. 6b). Moreover, the excellent chemo-selectivity of this process was due to other aromatic amino acid residue that were less electron-rich, so that the radical addition would not be favored. The modification of a variety of unprotected peptides was achieved effectively in this report with excellent conversions, and all of these polypeptides and proteins could be successfully and rapidly labeled with PTZ at room temperature. In particular, this protocol provides direct access to the late-stage derivatization of valuable drugs, such as

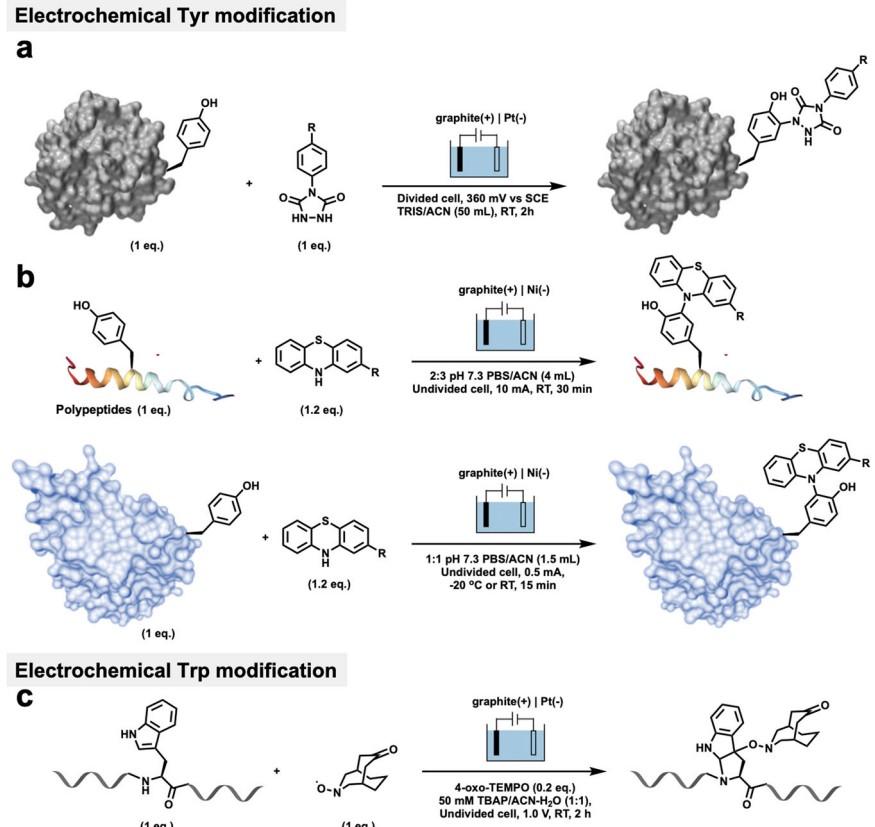

**Fig. 6 Recent electrochemical bioconjugations. a** The electrochemically promoted tyrosine-click reaction of peptides and proteins with urazoles, reported by Gouin and Boujtita in 2018 (ref. [75]). **b** Electrochemical bioconjugation of tyrosine-containing biomolecules with phenothiazine, reported by Lei and Chiang in 2019 (ref. [77]). **c** Electrochemical bioconjugation for labeling tryptophan by keto-ABNO, reported by Kanai and Oisaki in 2019 (ref. [78]).

the modification of uricosuric drugs with biotin, which can then be used to react with the pentapeptide YAGFL.

In this case, the author also applied this method to modifying proteins, such as insulin and myoglobin. When an excess amount of PTZ was added to insulin under electrolysis conditions, all of the reactive tyrosine residues were modified. Furthermore, myoglobin, a larger protein, was modified by an electro-oxidative bioconjugation method could be easily accomplished within 15 min, with a single PTZ being attached to myoglobin without having any influence on its structure. Therefore, this protocol would complement the current arsenal of techniques for the traceless preparation of a wide range of peptide and protein conjugates.

**Tryptophan**. Similar to the photocatalytic bioconjugation of tyrosine, electrochemical synthesis for selectively modifying tryptophan residues is also very important. In addition, with the environmentally friendly benefit of electrochemical synthesis, the merging of electrochemical synthesis and bioconjugation chemistry for modifying tryptophan is an attractive approach toward achieving these aims. However, reports of electrochemical bioconjugation for the modification of tryptophan are rare.

Kanai, Oisaki, and coworkers recently developed an electrochemical bioconjugation method for labeling tryptophan residues in peptides and proteins[78]. In comparison with the above electrochemical bioconjugation cases, this report showed that the electrochemical modification method could be applied to tryptophan, thus confirming the potential of electrochemical bioconjugation for use in modifying aromatic amino acids. This protocol demonstrated that two radicals, keto-ABNO and 4-oxo-TEMPO, were critical for inhibiting both the anodic overoxidation of the products and cross reactivity (Fig. 6c). In contrast to their original report using metal-free chemical activation, this electrochemical approach proceeded at a neutral pH and was accompanied by remarkably few side reactions, thus demonstrating the clean and scalable syntheses of chemically modified biologics.

### General points related to photoredox and electrochemical bioconjugation

In recent years, several bioconjugation reagents have been reported that address the issues of selectivity, toxicity, and biocompatibility, and are used for modifying biologically active biomolecules. However, exploiting the SET-based methods might may allow the target molecule to bind directly to the native peptide/protein without introducing a spacer/linker molecule, and this represents more synthetic options for synthetic chemists, biologists, and chemical biologists.

According to Table 1, there appear to be significant advantages associated with photoredox bioconjugation, since they typically proceed under mild conditions and visible light irradiation, as well as having excellent biocompatibility. However, the main limitation of photoredox bioconjugations is the longer reaction times (12–36 h) and the frequent need to exploit expensive noble metal photocatalysts. In addition, in some cases involving relatively poor selectivity, side reactions, such as overoxidation of Cys[21], excess treatment of oxidants or acids might cause side products[38,60], and the remaining fragment of substrates are all problems[47,54]. In addition, in cases of reactions that exploit transition metal-based photocatalysts the bioconjugates might need to be purified after the bioconjugation reactions, using a selectively permeable membrane. Although photoredox bioconjugation reactions have their disadvantages, this technique still be mild and straightforward processes for enriching the synthetic toolbox of bioconjugation.

On the other hand, electrochemical bioconjugations possess the benefits of relative rapid reaction times, controllable selectivity, as well as clean, mild, and scalable synthesis conditions. This would provide more opportunities for achieving biocompatible bioconjugation reactions in a more direct way. However, the unfamiliar operation processes of electrochemical syntheses for users, and the lack of electrochemical syntheses libraries each amino acid residue, will make the bio-applications of electrochemical bioconjugation uncertain.

### Application of SET-based peptide/protein bioconjugations in chemical biology

The SET-based bioconjugation strategy requires the integration of one reactive component into target biomolecules within cells or organisms. Proteins, peptides, and antibodies can be decorated with bioconjugation functional groups in a selective manner. As mentioned above, the advantages of the photoredox and electrochemical approaches include the possibility that the operating potential of the reaction can be directly and easily controlled by the simply external energy input. By introducing suitable labeling reagents, the amino acid residues of the proteins or peptides would then be attached in a straightforward manner, with a bioconjugate being produced with better selectivity.

However, since the reports of the photoredox and electrochemical bioconjugation sparse, there are only a few cases that are relevant to applications in biology. For example, SET-based peptide stapling and peptide macrocyclization represent a novel strategy for constraining peptides into organized macrocycles by two amino acids of a peptide. The resulting stapled peptides would efficiently bind to protein targets that are involved protein–protein interactions, thus improving the pharmacological properties[21,35,59,70].

Moreover, current SET-based bioconjugation reactions on proteins, such as the modification of insulin, myoglobin, and lysozyme promise to contribute to the possibility of their future in vitro or in vivo pharmaceutical applications[47,69,70,75,77].

In 2016, Jia reported on a strategy for the photoinduced covalent immobilization of proteins on phenol-functionalized surfaces. Through a ruthenium-catalyzed radical cross-linking reaction between proteins and phenol-modified surfaces, protein immobilization could be achieved. Moreover, the process is sufficiently mild that it can be applied to ar lipase, *Staphylococcus aureus* protein A, and streptavidin, with their bioactivity being preserved[79].

Nakamura and Sato recently reported on an electrochemically activated SET-based tyrosine modification that can bind N-methylated luminol on tyrosine residues of proteins and antibodies. By exploiting this electrochemical bioconjugation, they reported on the HC–Tyr57 site-specific modification of trastuzumab without gene manipulation. It should also be noted that the modification did not cause a fatal decrease in antigen binding affinity, and HC-Tyr57 site modified trastuzumab could be applied to in vivo imaging and ADC. The tumor imaging and antitumor activity of modified trastuzumab in NCI-N87 xenograft tumor model was investigated in this report[76].

Therefore, the straightforward control of external energy input SET-based techniques over bioconjugation reactions is one of the most attractive features.

### Outlook

The innovation of traditional bioconjugation and biorthogonal chemistry can be achieved through a SET strategy. By introducing photoredox catalysis and electrochemical synthesis techniques, several novel modifications of amino acids, peptides, and proteins can be accomplished. As shown by the examples herein,

**Table 1 Advantages and limitations of photoredox and electrochemical bioconjugations.**

| Amino acid residues | SET-based techniques | Bioconjugation targets | Advantages | Limitations | Reference |
|---|---|---|---|---|---|
| **Tyr** | Photoredox bioconjugation | —CF₃ | * high Tyr selectivity<br>* biocompatible conditions<br>* compatible with large peptides | * excess NaSO₂CF₃ needed<br>* oxidant or Ir-photocatalyst needed<br>* longer reaction times | Ref. 38 |
| | | | * rapid reactions<br>* aqueous media<br>* high yields of cross-linked products | * free Cys may be oxidized<br>* excess of photocatalyst needed<br>* used ammonium persulfate | Ref. 35 |
| | | | * trace-less photocatalyst<br>* aqueous media<br>* good target protein identity<br>* high Tyr selectivity<br>* aqueous media, mild conditions | * modified Ru-photocatalyst needed<br>* expensive | Ref. 37 |
| | Electrochemical bioconjugation | | * aqueous buffer, mild conditions<br>* no chemical oxidant<br>* scale covers peptides and proteins<br>* azide-armed peptides can be achieved | * sustainability of urazole synthesis<br>* side-reactions with Lys | Ref. 75 |
| | | | * high Tyr selectivity<br>* mild conditions, low controlled currents<br>* compatible with peptides and proteins<br>* short reaction times<br>* installation of high-value handles | * aqueous/organic co-solvent<br>* further modification of phenothiazines | Ref. 77 |
| **Trp** | Photoredox bioconjugation | —CF₃ | * less NaSO₂CF₃ needed<br>* mild conditions<br>* good Trp selectivity<br>* compatible with peptides | * Ir-photocatalyst needed<br>* aqueous/organic co-solvents | Ref. 45 |
| | | —(CF₂)₄F | * compatible with peptides<br>* no photocatalyst<br>* could proceed with sunlight | * organic solvents<br>* longer reaction times<br>* protein compatibility | Ref. 44 |
| | | | * no catalyst<br>* aqueous media<br>* installation of high-value handles<br>* compatible with peptides and proteins | * N-N cleavage by-products of pyridinium<br>* additional synthesis of pyridinium salts | Ref. 47 |
| | | | * good Trp selectivity<br>* installation of high-value handles<br>* compatible with peptides | * Ir-photocatalyst needed<br>* aqueous/organic co-solvents | Ref. 49 |
| | Electrochemical bioconjugation | | * mild conditions<br>* clean and scalable syntheses<br>* compatible with peptides and proteins<br>* good Trp selectivity | * aqueous/organic co-solvents<br>* reacts with amide bond of backbone | Ref. 78 |
| **His** | Photoredox bioconjugation | —CF₃ | * no photocatalyst<br>* *hv* needed<br>* compatible with oligo-peptides | * protein compatibility<br>* organic solvents and bases needed | Ref. 53 |
| | | | * high chemoselectivity<br>* compatible with peptides and proteins<br>* transition-metal free<br>* diverse R groups | * generation of by-products<br>* excess of dihydropyridine needed | Ref. 54 |
| | Electrochemical bioconjugation | - | - | - | - |
| **Cys** | Photoredox bioconjugation | | * mild conditions<br>* aqueous media<br>* compatible with peptides | * protein compatibility<br>* Ru-photocatalyst needed | Ref. 57 |
| | | | * high tolerance of sensitive amino acids<br>* aqueous media and mild conditions<br>* transition-metal free<br>* demonstrated use in stapling<br>* acceleration in continuous-flow | * protein compatibility<br>* lacks further modifications | Ref. 58 |
| | | | * metal-free photocatalyst<br>* compatible with peptides (PBS system)<br>* diverse R groups<br>* dinitrogen gas evolution | * organic solvents used in model reactions<br>* t-BuONO and TsOH needed | Ref. 60 |
| | | | * diverse R groups<br>* compatible with peptides<br>* 1°, 2° and 3° thiols compatible<br>* dual catalyst system | * organic solvents<br>* longer reaction times | Ref. 63 |
| | Electrochemical bioconjugation | - | - | - | - |
| **Ser** | Photoredox bioconjugation | | * Ni/photoredox dual catalysis system<br>* chemoselective C–O coupling<br>* compatible with macrocyclic peptides<br>* mild conditions | * longer reaction times<br>* aqueous/organic co-solvents | Ref. 67 |
| | | | * compatible with oligo-peptides<br>* mild conditions | * longer reaction times<br>* organic solvents | Ref. 68 |
| | Electrochemical bioconjugation | - | - | - | - |

photocatalytic and electrochemical approaches offer opportunities for carrying out diverse modifications of amino acids under mild, nontoxic, biocompatible, and environment friendly conditions. In addition, based on the reactivity of these SET-based bioconjugation processes in comparison to the conventional activation procedures. The bioorthogonal chemistry of the highly reactive Cys, Tyr, and Lys can be expected to be achieved in future applications. Therefore, photoredox and electrochemical catalysis can show new directions that clearly have benefits in that they have the potential to can enrich the toolbox of biomolecule chemical modifications.

While the development of photocatalysis and advances in electrosynthesis chemistry has progressed in this past decade, applications related to the modification of biomolecules continue to be rare. Therefore, more developments will be needed before photo-/electrochemical methodologies for protein modifications become commonplace. We anticipate that various strategies will be established that will permit the labeling of proteins.

To achieve this progress, the field of photo-/electrochemical bioconjugation will need more breakthroughs, especially for:

(i)   Broadly applicable site-selective transformations.
(ii)  Applications to complex peptides, proteins, and antibodies.
(iii) The use of a combination of photoredox/electrochemical/ transition metal catalysis methodologies, the development of dual catalysis to develop novel methods for selectively modifying aliphatic amino acid residues will be needed.
(iv)  Multicomponent bioorthogonal reaction processes that proceed by employing photo-/electrochemical strategies, giving access to more complex, multifunction or drug-like scaffolds will be needed.
(v)   More in-depth mechanistic studies.

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

## Acknowledgements

We wish to acknowledge the National Natural Science Foundation of China (grants 21520102003, 21701127, and 21702150). The Program of Introducing Talents of Discipline to Universities of China (111 Program) is also appreciated.

## Author contributions

Y.W., C.S., and C.-W.C. conducted the literature search. Y.W. and C.-W.C. contributed in writing the majority of the main text. Y.W., C.-W.C., and A. L. facilitated in the writing and editing of the main text.

## Competing interests

The authors declare no competing interests.
