## [Peer Review file · Communications Chemistry]

Reviewers' comments:

Reviewer #1 (Remarks to the Author):

The Perspective manuscript submitted by Aiwen Lei and co-workers, describing case studies on the chemical modification of peptides/proteins under photochemical/electrochemical conditions, has been reviewed. This Perspective highlighted the importance of single-electron transfer (SET) process as a common benefit for protein conjugations.

The Perspective in Communications Chemistry should be distinguished from a review article in terms proposing ideas and new future insights. The following major revision is recommended to make this manuscript fit better into the aim.

[General]

* At present, there are three reviews of photoredox/electrochemical peptide/protein modifications.

Noel, CEJ 2019, 25, 26. (ref 32)

Shatskiy, Matsuura, Karkas, Synthesis 2019, 51, 2759. (note cited)

Sato & Nakamura, Molecules 2019, 24, 3980. (not cited)

These reviews were published in very recent dates each other, and the cases discussed in this paper are inevitably very similar or overlapping the scope of these reviews. The current writing style seems to be just a coverage of these existing review articles together and add a few recent examples.

Although the authors were trying to extract commonality (SET process) as benefits in such conjugations, they can only claim not far from the benefits of photoredox catalysis. On the other hand, if we include the SET process by chemical activation, the scope is too huge to cover all of it in the limited length, I also understand this kind of scoping is essential.

In order to publish this Perspective, at least a differentiation from above reviews is required. In addition to regard single-electron transfer as a common benefit, I believe that "external energy input" is an important feature that characterizes the scope. Moreover, exemplified protein modifications in this Perspective are not always related to catalysis.

I would like to propose changing the title to clarify these points. How about the following title?

"SET-based peptide/protein bioconjugations driven by biocompatible energy input"

Based on this conceptual view, the authors can position visible light and electricity as biocompatible energy inputs by adding solid text, which will show new values to readers. Moreover, this conceptual view will help the readers' imagination toward not only in vitro level but also higher-order level such as application to cellular and animal systems. This will be preferred for fitting into the aim of Perspective.

* Using "biomolecule" in the title/text is clearly overclaiming because this Perspective only discuss protein/peptide modifications. "Biomolecule" includes sugar, nucleic acid, lipid, and so on, which would be rather misleading to readers. The word "biomolecule" should be avoided and substituted to "peptide/protein".

* Under most of presented chemical conditions, there are inherent restrictions on the amino acid compatibility and/or necessity of protecting groups. For example, Ser/Thr modifications usually requires rather strong conditions, which may incompatible with Cys, Tyr, Trp, and Met. The author must be recommend to comment and summarize such limitations by making tables or figures, in order to help readers' understanding.

* I personally recommend the authors to limit the scope of this Perspective within unmodified native amino acids. For example, the redox active ester (ref 61, 62) relies on prefunctionalization protocols and is never practical for expansion to the protein level. If such prefunctionalization examples are acceptable, photo-decaging and photoredox dehydroalanine(Dha)-modification are seemingly within the scope. This will blur the concept of this Perspective.

[1. Introduction]

* Although photoredox/electrochemical process shows different reactivity compared to conventional activation, orthogonality to highly reactive Cys is the most alarming aspect as well as the case of polar reactions. Please comment on this point to help readers seeing current issues and gaining a future perspective.

[2. Photoredox Bioconjugation]

[Trp]

* Please cite this newly published photochemical conditions applicable to proteins.
Taylor, JACS 2020, 142, 9112.

[Cys]

* I'm sure that current storyline must cover desulfurization as a useful conversion. Please cite these refs.

Guo, OL 2016, 18, 1166.

Cabrele & Reiser, J. Pept. Sci. 2017, 23, 556.

[His]

* Please create His section to include following refs describing photochemical conditions. If you want to cite ref47, please consider these as well.

Cohen, TL 1991, 31, 5705.

Chen & Wang, JACS 2019, 141, 18230.

[Ser/Thr]

* There is an example of photoredox transformation targeting homoserine-containing dipeptide.
Oisaki & Kanai, Synthesis 2020, DOI: 10.1055/s-0040-1707114

[Met]

* Although the text says "no photocatalytic methodologies for methionine bioconjugation have been reported", in reality there is one case of C-H cyanation of Met-containing dipeptide. Please correct the description and cite this paper.

Oisaki & Kanai, CEJ 2018, 24, 8501.

[C-terminal decarboxylation]

* Please also cite this photochemical conditions applicable to peptides.

Opatz, JOC 2019, 84, 2379.

[Dha]

* If you intend to create Dha section, please consider citing the following papers, which include peptide modifications.

Jui, ACS Catal. 2018, 8, 9115.

Roelfes, CEJ 2018, 24, 11314.

Mancheno, ChemCatChem 2019, 11, 3797.

Schubert, JOC 2020, 85, 6225.

[3. Electrochemical Bioconjugation]

* The authors should not miss the important example below which was recently published.

Sato & Nakamura, Bioconjugate Chem. 2020, 31, 1417.

[Reference]

* Comprehensive reviews of protein modifications seem to be randomly numbered and cited (They are cited as ref 2-9, 15-16, 36). Please define solid criteria for citation of previously published reviews and re-organize this section based on specific rules.

* The authors are recommended to limit citations which are absolutely essential for the discussion and contextualization. There are many references that seems not to be directly relevant. For example, I think the following refs are not cited in the right place, or they must be exchanged to contextually more appropriate reviews.

ref 10,17,18 : Please consider deletion.

ref 21 : This should be treated in the same context as ref 11-12.

ref 25 : This is a highly specific case, please consider removal.

ref 29,30 : Please consider deletion because these are not stating light-triggered biomolecule modification and not contextually necessary.

ref 31 : Please remove as out of context.

ref 32 : Please consider to cite following reviews in the same context.

Shatskiy, Matsuura, Karkas, Synthesis 2019, 51, 2759.

Sato & Nakamura, Molecules 2019, 24, 3980.

Correa, Asian JOC 2020, 9, 898.

ref 33 : Please consider deletion.

ref 65 : This should be treated in the same context as ref 26-28.

Reviewer #2 (Remarks to the Author):

In this article, Lei, Chiang and co-authors have described the recent development in chemical modification of peptides and proteins mediated by photoredox and electrochemical catalysis. The use of photoredox and electrochemical catalysis for bioconjugation is a new area, and the examples

remain limited as described by the authors. The idea is inspired by the rapid development of the photoredox and electrochemical catalysis in organic synthesis. Through the single electron transfer (SET) mechanism, these two approaches may provide new reactions for peptide and protein modification. However, several issues need to be addressed.

1. This perspective is too specialized from the point of view of synthetic chemists. As the authors suggest the potential applications of photoredox and electrochemical catalysis for the selective modification of biomolecules in biological studies, the authors should include and balance the view points and concerns of the readers and potential users of the bioconjugation reactions. The perspective could be supplemented with a guideline to provide information for not only synthetic chemists but also the biologists and chemical biologists on how to use these photoredox and electrochemical catalysis for bioconjugation of different types of biomolecules. The authors should describe both the advantages and disadvantages on the use of photoredox and electrochemical reactions for bioconjugation.
2. The authors may show the readers how to conduct the photoredox and electrochemical reactions for bioconjugation with suggested experimental set up as the procedures are quite diverse in the area of synthetic chemistry to perform photoredox and electrochemical catalysis.
3. One concern is that the authors compared the current bioconjugation methods with the photoredox and electrochemical catalysis. They described that there are a lot of the issues from the current bioconjugation techniques such as their “selectivity, toxicity, and biocompatibility...”, but in recent years, especially for cysteine modification, a number of publications have been reported to solve these problems and some of the reagents have been used for biologically active biomolecules. They also stated that “in the field of synthetic chemistry, the application of single electron transfer (SET) strategy to construct new complicated molecule are more popular” as well as “the molecule generation method could provide a new breakthrough than the traditional molecular synthesis approach.” Indeed, the photoredox and electrochemical synthesis are becoming more and more popular but it didn’t mean that the conventional method has become a “sunset industry”. In addition, in most of the examples discussed in this manuscript, photosensitizers or reactive reagents (e.g. keto-ABNO) are still needed in the photo- or electrochemical methods. Therefore, the authors are suggested to describe the photoredox and electrochemical catalysis as new directions with pros and cons which can enrich the toolbox for biomolecule chemical modifications.
4. Besides, the photoredox and electrochemical reactions may also have side reaction products as reported in the literature. Therefore, the authors should point out the formation of side reaction products so that the readers and users could have a full picture on the photoredox and electrochemical catalysis that can help them to consider all the reaction parameters when setting up the reactions for their use.
5. The authors are suggested to highlight the examples of bioconjugates (if any) prepared by the photoredox and electrochemical catalysis in which the biological activities of these bioconjugates are comparable with that of the native biomolecules. Thus, the biologists and chemical biologists are willing to pay more attention to use the photoredox and electrochemical reactions for chemical modifications of biomolecules. For example, antibody-drug conjugates (ADCs) are very important bioconjugates in biopharmaceutical industry. Any literature example on the use of photoredox and electrochemical catalysis for modification of antibodies or biologically active proteins?
6. The authors should remind the readers and users that certain photocatalysts (especially transition metal-based catalysts) would potentially make the purification of bioconjugates more complicated after bioconjugation reactions, imparting toxicity, phototoxicity and biocidal activities in the

biological studies. Any advices could be given to the readers and users how to tackle and remove the photocatalysts and other excess reagents after the bioconjugations?

7. To encourage the readers to consider using photoredox and electrochemical reactions for bioconjugations, the authors may highlight some specific and successful examples (if any) that could only be achieved by using the photoredox catalysis and electrochemical catalysis rather than other approaches.

8. The English writing of this perspective needs to be significantly improved as there are a lot of grammatical and spelling mistakes, and some unclear expressions which would make the readers confused about their opinions. For example, in Line 8, "Bioconjugation and bioorthogonal reactions play a central facilitating role for engendering artificial peptides and proteins." The meaning of the "central facilitating role" is unclear and the "artificial peptides and proteins" may usually refer to synthetic peptides for chemists. Another example is from Line 43 to 49, "Although traditional reagents ... in the most of amino acids have been yet clearly exploited." This paragraph is describing the importance to access the site-selectivity in protein modification. But these sentences would make the readers feel unclear about what the authors want to express.

9. Another issue is that the authors should be very careful about the terms used in the manuscript that need to follow the general definition in the field. For example, they used the term "site selective" to describe the number of modifications in this review. However, for peptide and protein modifications, "site selective" usually means "a covalent bond between a protein and a synthetic organic molecule at a pre-defined residue" [Nat. Chem. 2016, 8, 103], which is different from "chemoselective". In this article, except for the C-terminal modification, it is hard to claim the other methods to be site selective. Moreover, in principle, the relationship between the site selectivity and the single electron transfer remains unclear and hence it is better not to include the site selectivity when the site selectivity is unknown. The term "biomolecules" used in the title is too broad as it involves not only the modification of peptides and proteins but also modifications of polynucleotide chains (DNA and RNA) as well as lipids and sugars. Similar case is the term "bioorthogonal" which has been described in the abstract and conclusion, but no reaction has been mentioned in the main text. There is also a number of terms like "bio-conjugal", "active reagents", "inert amino acid", "site-chain" which should be revised.

10. Some mistakes are found in the schemes. For example, in Scheme 2a, the perfluoroalkyl should be $(CF_2)_4F$. In Scheme 2b, the product depicted was not the correct product. In Scheme 2c, the reaction intermediate did not contain CF_3 in the structure.

Reviewer #3 (Remarks to the Author):

The perspective by Lei and co-workers describes the selective modification of biomolecules using photoredox and electrochemical catalysis. This is an important emerging field deserving a specific discussion. This short review presents important examples in the field but significant studies are still missing. Importantly the grammatical English of the manuscript is poor and nearly all the sections are difficult to follow and understand (i.e. Line 11-14, Line 191 a 198, line 238-241, line 258-262, line 269-273, line 282- 285, line 372-374....). The introduction and conclusion do not deliver a clear message. What is the complementary potential of photoredox and electrochemical methods over more conventional Lys and Cys modifications? The schemes are never cited in the text, and the experimental methods (that are only superficially described) are difficult to follow. Mechanisms of SET and electrochemistry should be described, please note that electrochemical oxidation is not

necessary a SET mechanisms (as stated line 11). My opinion is that the manuscript is not acceptable at this stage of maturation and I recommend to the author to remove the broad statements and to carefully improve the English and discussion prior resubmission in another journal.

The manuscript has been carefully revised and modified according to the comments. Below are our point-to-point answers and statements to all the suggestions and comments.

Reviewer #1 (Remarks to the Author):

The Perspective manuscript submitted by Aiwen Lei and co-workers, describing case studies on the chemical modification of peptides/proteins under photochemical/electrochemical conditions, has been reviewed. This Perspective highlighted the importance of single-electron transfer (SET) process as a common benefit for protein conjugations.

The Perspective in Communications Chemistry should be distinguished from a review article in terms proposing ideas and new future insights. The following major revision is recommended to make this manuscript fit better into the aim.

[General]

* At present, there are three reviews of photoredox/electrochemical peptide/protein modifications.

Noel, CEJ 2019, 25, 26. (ref 32)

Shatskiy, Matsuura, Karkas, Synthesis 2019, 51, 2759. (note cited)

Sato & Nakamura, Molecules 2019, 24, 3980. (not cited)

These reviews were published in very recent dates each other, and the cases discussed in this paper are inevitably very similar or overlapping the scope of these reviews. The current writing style seems to be just a coverage of these existing review articles together and add a few recent examples.

Although the authors were trying to extract commonality (SET process) as benefits in such conjugations, they can only claim not far from the benefits of photoredox catalysis. On the other hand, if we include the SET process by chemical activation, the scope is too huge to cover all of it in the limited length, I also understand this kind of scoping is essential.

Our response: Thanks for your suggestions. Indeed, recently there are some similar review articles have been published in other journals. Such as CEJ 2019 (photoredox bioconjugations), Synthesis 2019 (photoredox bioconjugations), and Molecules 2019 (radical chemistry for bioconjugations).

However, in this Perspective, we not only would like to incorporate the reports of the previous relative works but also want to emphasize the advantages of SET-based bioconjugation chemistry and their future utilities on chemical biology. As we mentioned in the manuscript, reports of photoredox/electrochemical bioconjugations are still significantly limited, especially for electrochemical bioconjugations. The electrochemical synthesis for C-H bond activation was just stepwise developed from

2017. However, the field of SET-based bioconjugation still needs more breakthroughs. After the analysis of this Perspective, we hope that researchers could turn their focus to the environmentally friendly electrochemical bioconjugations as well as visible-light photoredox bioconjugations. By using these straightforward and biocompatible methods, researchers could broaden the applications of late-stage functionalizations on biological systems.

In order to publish this Perspective, at least a differentiation from above reviews is required. In addition to regard single-electron transfer as a common benefit, I believe that "external energy input" is an important feature that characterizes the scope. Moreover, exemplified protein modifications in this Perspective are not always related to catalysis.

I would like to propose changing the title to clarify these points. How about the following title?

"SET-based peptide/protein bioconjugations driven by biocompatible energy input"

Based on this conceptual view, the authors can position visible light and electricity as biocompatible energy inputs by adding solid text, which will show new values to readers. Moreover, this conceptual view will help the readers' imagination toward not only in vitro level but also higher-order level such as application to cellular and animal systems. This will be preferred for fitting into the aim of Perspective.

Our response: Thanks for your suggestions. This title seems great, and we have changed the title as your suggestion. Besides, we also added some references and descriptions for the biological applications of the SET-based bioconjugations.

* Using "biomolecule" in the title/text is clearly overclaiming because this Perspective only discuss protein/peptide modifications. "Biomolecule" includes sugar, nucleic acid, lipid, and so on, which would be rather misleading to readers. The word "biomolecule" should be avoided and substituted to "peptide/protein".

Our response: Thank you for your suggestion. We have adjusted the description in the revised manuscript.

* Under most of presented chemical conditions, there are inherent restrictions on the amino acid compatibility and/or necessity of protecting groups. For example, Ser/Thr modifications usually requires rather strong conditions, which may incompatible with Cys, Tyr, Trp, and Met. The author must be recommend to comment and summarize such limitations by making tables or figures, in order to help

readers' understanding.

Our response: Thank you for your suggestion. We have added a table which groups photoredox/electrochemical bioconjugation methods targeting each amino acid residue and summarises their advantages and disadvantages.

* I personally recommend the authors to limit the scope of this Perspective within unmodified native amino acids. For example, the redox active ester (ref 61, 62) relies on prefunctionalization protocols and is never practical for expansion to the protein level. If such prefunctionalization examples are acceptable, photo-decaging and photoredox dehydroalanine(Dha)-modification are seemingly within the scope. This will blur the concept of this Perspective.

Our response: Thanks for the suggestion. We have adjusted the description in the revised manuscript, and we removed the pre-functionalization examples in the main text.

[1. Introduction]

* Although photoredox/electrochemical process shows different reactivity compared to conventional activation, orthogonality to highly reactive Cys is the most alarming aspect as well as the case of polar reactions. Please comment on this point to help readers seeing current issues and gaining a future perspective.

Our response: Thanks for the suggestion. We have adjusted the description and provided our comment on this point in the revised manuscript.

[2. Photoredox Bioconjugation]

[Trp]

* Please cite this newly published photochemical conditions applicable to proteins. Taylor, JACS 2020, 142, 9112.

Our response: Thanks for the suggestion. We have cited this paper and described its importance in the revised manuscript.

[Cys]

* I'm sure that current storyline must cover desulfurization as a useful conversion. Please cite these refs.

Guo, OL 2016, 18, 1166.

Cabrele & Reiser, J. Pept. Sci. 2017, 23, 556.

Our response: Thanks for the suggestion. We have cited these papers and described their importance in the revised manuscript.

[His]

* Please create His section to include following refs describing photochemical conditions. If you want to cite ref47, please consider these as well.

Cohen, TL 1991, 31, 5705.

Chen & Wang, JACS 2019, 141, 18230.

Our response: Thanks for the suggestion. We have cited these papers and created His section to emphasize their importance in the revised manuscript.

[Ser/Thr]

* There is an example of photoredox transformation targeting homoserine-containing dipeptide.

Oisaki & Kanai, Synthesis 2020, DOI: 10.1055/s-0040-1707114

Our response: Thanks for the suggestion. We have cited this paper and described its importance in the revised manuscript.

[Met]

* Although the text says "no photocatalytic methodologies for methionine bioconjugation have been reported", in reality there is one case of C-H cyanation of Met-containing dipeptide. Please correct the description and cite this paper.

Oisaki & Kanai, CEJ 2018, 24, 8501.

Our response: Thanks for the suggestion. We have cited this paper, modified the description of the Met, and reported its importance in the revised manuscript.

[C-terminal decarboxylation]

* Please also cite this photochemical conditions applicable to peptides.

Opatz, JOC 2019, 84, 2379.

Our response: Thanks for the suggestion. We have cited this paper and described its content in the revised manuscript.

[Dha]

* If you intend to create Dha section, please consider citing the following papers, which include peptide modifications.

Jui, ACS Catal. 2018, 8, 9115.

Roelfes, CEJ 2018, 24, 11314.

Mancheno, ChemCatChem 2019, 11, 3797.

Schubert, JOC 2020, 85, 6225.

Our response: Thanks for your suggestion. However, we would not plan to create the

Dha section because we feel this part would be out of the scope and space limitation.

[3. Electrochemical Bioconjugation]

* The authors should not miss the important example below which was recently published.

Sato & Nakamura, *Bioconjugate Chem.* 2020, 31, 1417.

Our response: Thanks for the suggestion. We have cited this paper and described its content in the revised manuscript.

[Reference]

* Comprehensive reviews of protein modifications seem to be randomly numbered and cited (They are cited as ref 2-9, 15-16, 36). Please define solid criteria for citation of previously published reviews and re-organize this section based on specific rules.

Our response: Thanks for the suggestion. We have modified this issue in the revised manuscript.

* The authors are recommended to limit citations which are absolutely essential for the discussion and contextualization. There are many references that seems not to be directly relevant. For example, I think the following refs are not cited in the right place, or they must be exchanged to contextually more appropriate reviews.

ref 10,17,18 : Please consider deletion.

ref 21 : This should be treated in the same context as ref 11-12.

ref 25 : This is a highly specific case, please consider removal.

ref 29,30 : Please consider deletion because these are not stating light-triggered biomolecule modification and not contextually necessary.

ref 31 : Please remove as out of context.

ref 32 : Please consider to cite following reviews in the same context.

Shatskiy, Matsuura, Karkas, *Synthesis* 2019, 51, 2759.

Sato & Nakamura, *Molecules* 2019, 24, 3980.

Correa, *Asian JOC* 2020, 9, 898.

ref 33 : Please consider deletion.

ref 65 : This should be treated in the same context as ref 26-28.

Our response: Thanks for the suggestion. We have modified references as you mentioned in the revised manuscript.

Reviewer #2 (Remarks to the Author):

In this article, Lei, Chiang and co-authors have described the recent development in chemical modification of peptides and proteins mediated by photoredox and electrochemical catalysis. The use of photoredox and electrochemical catalysis for bioconjugation is a new area, and the examples remain limited as described by the authors. The idea is inspired by the rapid development of the photoredox and electrochemical catalysis in organic synthesis. Through the single electron transfer (SET) mechanism, these two approaches may provide new reactions for peptide and protein modification. However, several issues need to be addressed.

1. This perspective is too specialized from the point of view of synthetic chemists. As the authors suggest the potential applications of photoredox and electrochemical catalysis for the selective modification of biomolecules in biological studies, the authors should include and balance the view points and concerns of the readers and potential users of the bioconjugation reactions. The perspective could be supplemented with a guideline to provide information for not only synthetic chemists but also the biologists and chemical biologists on how to use these photoredox and electrochemical catalysis for bioconjugation of different types of biomolecules. The authors should describe both the advantages and disadvantages on the use of photoredox and electrochemical reactions for bioconjugation.

Our response: Thanks for the suggestion. In the revised version, we added a table which groups methods targeting each amino acid residue and summarises their advantages and disadvantages. Besides, we added a section of biological applications for broadening the value of the readers beyond synthetic organic chemists.

2. The authors may show the readers how to conduct the photoredox and electrochemical reactions for bioconjugation with suggested experimental set up as the procedures are quite diverse in the area of synthetic chemistry to perform photoredox and electrochemical catalysis.

Our response: Thanks for the suggestion. We think a detailed description of experimental set-ups would be out of scope, but we still tried to increase some brief descriptions of the additional practical set-ups in the revised manuscript.

3. One concern is that the authors compared the current bioconjugation methods with the photoredox and electrochemical catalysis. They described that there are a lot of the issues from the current bioconjugation techniques such as their “selectivity, toxicity, and biocompatibility...”, but in recent years, especially for cysteine

modification, a number of publications have been reported to solve these problems and some of the reagents have been used for biologically active biomolecules. They also stated that “in the field of synthetic chemistry, the application of single electron transfer (SET) strategy to construct new complicated molecule are more popular” as well as “the molecule generation method could provide a new breakthrough than the traditional molecular synthesis approach.” Indeed, the photoredox and electrochemical synthesis are becoming more and more popular but it didn’t mean that the conventional method has become a “sunset industry”. In addition, in most of the examples discussed in this manuscript, photosensitizers or reactive reagents (e.g. keto-ABNO) are still needed in the photo- or electrochemical methods. Therefore, the authors are suggested to describe the photoredox and electrochemical catalysis as new directions with pros and cons which can enrich the toolbox for biomolecule chemical modifications.

Our response: Thanks for the suggestion. Of course, we did not mention that the conventional method has become a sunset industry. To date, the major bioconjugation strategy is still focused on developing new bioconjugation/bioorthogonal reagents, and indeed, these research works are essential to improve the applications of chemical biology. In this Perspective, we intend to emphasize the advantages of SET-based bioconjugation chemistry and their future utilities on chemical biology. Besides, we want to deliver a concept to readers that photoredox/electrochemical bioconjugations might provide opportunities to make the combination of an unreactive substrate with target amino acid residue. Moreover, as your comment, we have added a section and a table for summarising the advantages and limitations of photoredox/electrochemical bioconjugations in the revised manuscript.

4. Besides, the photoredox and electrochemical reactions may also have side reaction products as reported in the literature. Therefore, the authors should point out the formation of side reaction products so that the readers and users could have a full picture on the photoredox and electrochemical catalysis that can help them to consider all the reaction parameters when setting up the reactions for their use.

Our response: Thanks for the suggestion. We have pointed out some possible side reactions in the revised manuscript, such as the over oxidation of Cys (ref. 35) and Lys (ref. 75) or the N-N bond cleavage side product of pyridinium salts (ref. 47). Unfortunately, most of the reports did not provide the data of side products, and we are hard to describe the side products in each reaction. In general, side products will be purified through filtration, dialysis, or chromatography in these works.

5. The authors are suggested to highlight the examples of bioconjugates (if any)

prepared by the photoredox and electrochemical catalysis in which the biological activities of these bioconjugates are comparable with that of the native biomolecules. Thus, the biologists and chemical biologists are willing to pay more attention to use the photoredox and electrochemical reactions for chemical modifications of biomolecules. For example, antibody-drug conjugates (ADCs) are very important bioconjugates in biopharmaceutical industry. Any literature example on the use of photoredox and electrochemical catalysis for modification of antibodies or biologically active proteins?

Our response: Thanks for the suggestion. We think the bioconjugates themselves prepared by the photoredox and electrochemical catalysis might be out of the scope of this Perspective. However, in the revised version, we added a section for the biological applications of photoredox/electrochemical bioconjugations, which describe their application on antibody-drug conjugates (ADCs).

6. The authors should remind the readers and users that certain photocatalysts (especially transition metal-based catalysts) would potentially make the purification of bioconjugates more complicated after bioconjugation reactions, imparting toxicity, phototoxicity and biocidal activities in the biological studies. Any advices could be given to the readers and users how to tackle and remove the photocatalysts and other excess reagents after the bioconjugations?

Our response: Thanks for the suggestion. We have modified the description in the revised manuscript. Side products and catalysts generally could be purified through filtration, dialysis, or chromatography after bioconjugation reactions.

7. To encourage the readers to consider using photoredox and electrochemical reactions for bioconjugations, the authors may highlight some specific and successful examples (if any) that could only be achieved by using the photoredox catalysis and electrochemical catalysis rather than other approaches.

Our response: Thanks for the suggestion. About your concern, indeed, there are some cases such as trifluoromethylation and Tyr-Tyr coupling could be achieved by other methods. However, there are still several approaches that have to use the SET-based methods, i.e. photoredox or electrochemical catalysis, to proceed with the reaction. For example, ref. 44, 47, 54, 60, 68, 77 and 78, almost every case is unreactive by previous reaction conditions. Therefore, it is hard to highlight each approach that could only be performed by the SET-based methods. However, as your concern, we still tried to emphasize some individual cases in the revised manuscript.

8. The English writing of this perspective needs to be significantly improved as

there are a lot of grammatical and spelling mistakes, and some unclear expressions which would make the readers confused about their opinions. For example, in Line 8, “Bioconjugation and bioorthogonal reactions play a central facilitating role for engendering artificial peptides and proteins.” The meaning of the “central facilitating role” is unclear and the “artificial peptides and proteins” may usually refer to synthetic peptides for chemists. Another example is from Line 43 to 49, “Although traditional reagents ... in the most of amino acids have been yet clearly exploited.” This paragraph is describing the importance to access the site-selectivity in protein modification. But these sentences would make the readers feel unclear about what the authors want to express.

Our response: Thanks for the suggestion. The text is carefully examined for English usage and scientific content in the revised version.

9. Another issue is that the authors should be very careful about the terms used in the manuscript that need to follow the general definition in the field. For example, they used the term “site selective” to describe the number of modifications in this review. However, for peptide and protein modifications, “site selective” usually means “a covalent bond between a protein and a synthetic organic molecule at a pre-defined residue” [Nat. Chem. 2016, 8, 103], which is different from “chemoselective”. In this article, except for the C-terminal modification, it is hard to claim the other methods to be site selective. Moreover, in principle, the relationship between the site selectivity and the single electron transfer remains unclear and hence it is better not to include the site selectivity when the site selectivity is unknown. The term “biomolecules” used in the title is too broad as it involves not only the modification of peptides and proteins but also modifications of polynucleotide chains (DNA and RNA) as well as lipids and sugars. Similar case is the term “bioorthogonal” which has been described in the abstract and conclusion, but no reaction has been mentioned in the main text. There is also a number of terms like “bio-conjugal”, “active reagents”, “inert amino acid”, “site-chain” which should be revised.

Our response: Thanks for the suggestion. The terms have been modified in the revised version.

10. Some mistakes are found in the schemes. For example, in Scheme 2a, the perfluoroalkyl should be $(CF_2)_4F$. In Scheme 2b, the product depicted was not the correct product. In Scheme 2c, the reaction intermediate did not contain CF_3 in the structure

Our response: Thanks for the suggestion. The schemes have been modified in the revised version.

Reviewer #3 (Remarks to the Author):

The perspective by Lei and co-workers describes the selective modification of biomolecules using photoredox and electrochemical catalysis. This is an important emerging field deserving a specific discussion. This short review presents important examples in the field but significant studies are still missing. Importantly the grammatical English of the manuscript is poor and nearly all the sections are difficult to follow and understand (i.e. Line 11-14, Line 191 a 198, line 238-241, line 258-262, line 269-273, line 282- 285, line 372-374....). The introduction and conclusion do not deliver a clear message. What is the complementary potential of photoredox and electrochemical methods over more conventional Lys and Cys modifications? The schemes are never cited in the text, and the experimental methods (that are only superficially described) are difficult to follow. Mechanisms of SET and electrochemistry should be described, please note that electrochemical oxidation is not necessary a SET mechanisms (as stated line 11). My opinion is that the manuscript is not acceptable at this stage of maturation and I recommend to the author to remove the broad statements and to carefully improve the English and discussion prior resubmission in another journal.

Our response: Thanks for the suggestion. In our opinion, the major bioconjugation strategy is still focused on developing new bioconjugation/bioorthogonal reagents, and indeed, these research works are essential to improve the applications of chemical biology.

In this Perspective, we intend to emphasize the advantages of SET-based bioconjugation chemistry and their future utilities on chemical biology. Besides, we want to deliver a concept to readers that photoredox/electrochemical bioconjugations might provide opportunities to make the combination of an unreactive substrate with target amino acid residue. In addition, we added a table which groups methods targeting each amino acid residue and summarises their advantages and disadvantages.

About the mechanism studies, the detailed discussion of mechanistic minutiae maybe beyond the scope of the piece, so we tried to give some brief discussions of general trends. Meanwhile, as your suggestion, we have carefully examined this manuscript for English usage and scientific content in the revised version.

REVIEWERS' COMMENTS:

Reviewer #1 (Remarks to the Author):

I reviewed the revised manuscript and feel satisfied with all the corrections. Please publish without further modifications.

Reviewer #2 (Remarks to the Author):

The authors have spent significant effort to improve the quality of this perspective. However, there remains some critical issues that needed to be addressed.

1. The manuscript is still not able to deliver a clear message to the target researchers. It would be better for the authors to present more advantages of photoredox and electrochemical methods over conventional Lys and Cys modifications. Moreover, how the photoredox and electrochemical approaches can be used as a complementary approach to conventional Lys and Cys modifications could be discussed.

2. Although photoredox catalysis and electrochemical synthesis are becoming more popular for synthetic chemists, as for chemical biologists, they are still regarded as new tools. The authors are suggested to prepare this manuscript for targeting not only synthetic chemists but also structural and chemical biologists. Therefore, to encourage these researchers to explore photoredox and electrochemical methods, a general guideline of the reaction conditions and experimental setup would be useful in this perspective, which has been suggested in previous comments. For example, for commonly used modification, mild reaction conditions (aqueous solvent, pH 6–8, temperature < 37 °C) are required, and nucleophilic residues (cysteine and lysine) are more reactive. The authors are suggested to provide important guidelines of the wavelength range, power source and voltage region for applying these new photoredox and electrochemical approaches to bioconjugation.

3. The revised Scheme 1, which is a general scheme for comparison between the traditional approaches to the photoredox and electrochemical modifications, contains critical misleading information. For example, they pointed out that traditional methods gave weak biocompatibility and selectivity. However, during the past decades, many chemoselective modifications have been developed and some of them have been applied for in vivo modifications. Besides, if the authors insist on comparing the biocompatibility among these methods, they may need to describe the cases following this theme. However, apart from some of the electrochemical methods, it is hard to justify the new approaches are more biocompatible than the traditional methods. In addition, some of the cases described in this perspective are not biocompatible (e.g. photoredox serine modification was conducted in organic solvents). The authors may consider to revise this claim. In addition, the controllable synthesis of electrochemical modification has not been pointed out in the text and it is suggested to delete this claim.

4. In the rebuttal letter, the authors claimed that “we want to deliver a concept to readers that photoredox/electrochemical bioconjugations might provide opportunities to make the combination of an unreactive substrate with target amino acid residue.” The concept is interesting for dual modification or photo-triggered selective modification. However, for general modification, there should be no benefit to use non-reactive reagents as reactive reagents that would give a shorter reaction time. In addition, for the cases discussed in this perspective, after activation by light or electricity, the unreactive reagents or the peptides would also become reactive to achieve high conversions.

5. After the revision, the perspective appears to be too long. Instead of describing introductions to different amino acids which have been described by previous reviews, they are suggested to compress those introductions. A more interesting point will be how different amino acids behave in SET reactions. They are also suggested to compress some early cases and focus on cases which are more relevant to the concepts that they want to share.

6. They authors are suggested to revise the section 2.1 as histidine modification has been incorporated into the text. Revision of all the schemes are also suggested as some the newly added cases are missing in those schemes.

7. In addition, the schemes are not cited properly in the text so we have difficulty to correlate the schemes and text. Thus, the authors are suggested to check them carefully.

8. Some spelling and grammatical errors are found. For example,

- a) "visible-light induced" and "visible-light-induced"
- b) Some odd codes appear like "dtb□-bpy", "sulfur□-containing"
- c) "Previousl studies"
- d) "...direct labelling of in-chain serine and threonine...", do you mean "side-chain"??
- e) "electro-chemical" and "electro-chemically" etc.

9 In C-terminal decarboxylative modifications, the author describes riboflavin tetrabutyrate was used as an efficient water-soluble organic photocatalyst, however, the scheme 5(b) did not show the correct structure of riboflavin tetrabutyrate.

Reviewer #3 (Remarks to the Author):

The revised version of the manuscript by Lei and co-workers is much more enjoyable to read and most of the comments raised by the referee have been carefully addressed.

In consequence I recommend publication after the following minor modifications.

Some symbols disappeared during pdf conversion (l 293, 298)

Table1 (Tyr): Side-reaction with Lys is not observed with the electrochemical process.

Reviewer #1 (Remarks to the Author):

I reviewed the revised manuscript and feel satisfied with all the corrections. Please publish without further modifications.

Our response: Thank you for your kind comment.

Reviewer #2 (Remarks to the Author):

The authors have spent significant effort to improve the quality of this perspective. However, there remains some critical issues that needed to be addressed.

1. The manuscript is still not able to deliver a clear message to the target researchers. It would be better for the authors to present more advantages of photoredox and electrochemical methods over conventional Lys and Cys modifications. Moreover, how the photoredox and electrochemical approaches can be used as a complementary approach to conventional Lys and Cys modifications could be discussed.

Our response: Thank you for your comment. According to your comment, the change in describing the complementary approach to conventional Lys and Cys modifications would be out of scope, so we think that is unnecessary at this stage.

2. Although photoredox catalysis and electrochemical synthesis are becoming more popular for synthetic chemists, as for chemical biologists, they are still regarded as new tools. The authors are suggested to prepare this manuscript for targeting not only synthetic chemists but also structural and chemical biologists. Therefore, to encourage these researchers to explore photoredox and electrochemical methods, a general guideline of the reaction conditions and experimental setup would be useful in this perspective, which has been suggested in previous comments. For example, for commonly used modification, mild reaction conditions (aqueous solvent, pH 6~8, temperature < 37 °C) are required, and nucleophilic residues (cysteine and lysine) are more reactive. The authors are suggested to provide important guidelines of the wavelength range, power source and voltage region for applying these new photoredox and electrochemical approaches to bioconjugation.

Our response: Thank you for your comment. We have modified all the schemes in the manuscript. The experimental parameters on each reaction scheme in the article, for

example, temperature, reaction time, solvent, catalyst/photosensitiser loading and wavelength/voltage have been added in the equations for audience's guidelines.

3. The revised Scheme 1, which is a general scheme for comparison between the traditional approaches to the photoredox and electrochemical modifications, contains critical misleading information. For example, they pointed out that traditional methods gave weak biocompatibility and selectivity. However, during the past decades, many chemoselective modifications have been developed and some of them have been applied for in vivo modifications. Besides, if the authors insist on comparing the biocompatibility among these methods, they may need to describe the cases following this theme. However, apart from some of the electrochemical methods, it is hard to justify the new approaches are more biocompatible than the traditional methods. In addition, some of the cases described in this perspective are not biocompatible (e.g. photoredox serine modification was conducted in organic solvents). The authors may consider to revise this claim. In addition, the controllable synthesis of electrochemical modification has not been pointed out in the text and it is suggested to delete this claim.

Our response: Thank you for your suggestion. We have adjusted the description in the revised manuscript.

4. In the rebuttal letter, the authors claimed that “we want to deliver a concept to readers that photoredox/electrochemical bioconjugations might provide opportunities to make the combination of an unreactive substrate with target amino acid residue.” The concept is interesting for dual modification or photo-triggered selective modification. However, for general modification, there should be no benefit to use non-reactive reagents as reactive reagents that would give a shorter reaction time. In addition, for the cases discussed in this perspective, after activation by light or electricity, the unreactive reagents or the peptides would also become reactive to achieve high conversions.

Our response: Thank you for your suggestion. Indeed, the unreactive reagents and peptides would become reactive under the reaction processes. However, to introduce and explore the “unreactive” reagents would provide the benefit for the modifications that would like to directly attach the drugs, bioactive species and probes without the pre-functionalisation.

5. After the revision, the perspective appears to be too long. Instead of describing

introductions to different amino acids which have been described by previous reviews, they are suggested to compress those introductions. A more interesting point will be how different amino acids behave in SET reactions. They are also suggested to compress some early cases and focus on cases which are more relevant to the concepts that they want to share.

Our response: Thank you for your suggestion. However, this comment implies substantial changes to the focus of the piece, and we do not believe that is necessary at this stage.

6. They authors are suggested to revise the section 2.1 as histidine modification has been incorporated into the text. Revision of all the schemes are also suggested as some the newly added cases are missing in those schemes.

Our response: Thank you for your suggestion. Indeed, in some schemes, the newly added cases should be incorporated into the figures. However, the additional instances mainly focus on the methodologies of simple molecule modifications and their further late-stage applications on peptides. Therefore, to prevent this perspective appears to be too long, we didn't add all the cases into the schemes.

7. In addition, the schemes are not cited properly in the text so we have difficulty to correlate the schemes and text. Thus, the authors are suggested to check them carefully.

Our response: Thank you for your suggestion. We have adjusted the schemes with references in the revised manuscript.

8. Some spelling and grammatical errors are found. For example,

a) “visible-light induced” and “visible-light-induced”

b) Some odd codes appear like “dtb□-bpy”, “sulfur□-containing”

c) “Previousl studies”

d) “...direct labelling of in-chain serine and threonine...”, do you mean “side-chain”??

e) “electrochemical” and “electro-chemically” etc.

Our response: Thank you for your suggestion. We have modified these errors in the revised manuscript. The problem of (b) should be some error happened while the transformation of pdf version and we have tried to correct it. Besides, the description “in-chain” of (d) means the serine in the peptide chain, to prevent the misunderstanding, we have changed the description and deleted the term of in-chain.

9 In C-terminal decarboxylative modifications, the author describes riboflavin tetrabutryate was used as an efficient water-soluble organic photocatalyst, however, the scheme 5(b) did not show the correct structure of riboflavin tetrabutryate.

Our response: Thank you for your suggestion. We have modified the structure in the revised manuscript.

Reviewer #3 (Remarks to the Author):

The revised version of the manuscript by Lei and co-workers is much more enjoyable to read and most of the comments raised by the referee have been carefully addressed.

In consequence I recommend publication after the following minor modifications.

Some symbols disappeared during pdf conversion (l 293, 298)

Table1 (Tyr): Side-reaction with Lys is not observed with the electrochemical process.

Our response: Thank you for your suggestion. We have adjusted the descriptions in the revised manuscript.